# $T_c$ and Other Cuprate Properties in Relation to Planar Charges as Measured by NMR

**Michael Jurkutat [1], Andreas Erb [2] and Jürgen Haase [1,*]** 

[1]   Felix Bloch Institute for Solid State Physics, University of Leipzig, Linnéstr. 5, 04103 Leipzig, Germany
[2]   Walther Meissner Institut, Bayerische Akademie der Wissenschaften, 85748 Garching, Germany
[*]   Correspondence: j.haase@physik.uni-leipzig.de; Tel.: +49-341-97-32601

**Abstract:** Nuclear magnetic resonance (NMR) in cuprate research is a prominent bulk local probe of magnetic properties. NMR also, as was shown over the last years, actually provides a quantitative measure of local charges in the $CuO_2$ plane. This has led to fundamental insights, e.g., that the maximum $T_c$ is determined by the sharing of the parent planar hole between Cu and O. Using bonding orbital hole contents on planar Cu and O measured by NMR, instead of the total doping $x$, the thus defined two-dimensional cuprate phase diagram reveals significant differences between the various cuprate materials. Even more importantly, the reflected differences in material chemistry appear to set a number of electronic properties as we discuss here, for undoped, underdoped and optimally doped cuprates. These relations should advise attempts at a theoretical understanding of cuprate physics as well as inspire material chemists towards new high-$T_c$ materials. Probing planar charges, NMR is also sensitive to charge variations or ordering phenomena in the $CuO_2$ plane. Thereby, local charge order on planar O in optimally doped YBCO could recently be proven. Charge density variations seen by NMR in both planar bonding orbitals with amplitudes between 1% to 5% appear to be omnipresent in the doped $CuO_2$ plane, i.e., not limited to underdoped cuprates and low temperatures.

**Keywords:** NMR; cuprates; charge order

## 1. Introduction

Nuclear magnetic resonance (NMR) as a bulk, local quantum sensor provides unparalleled insights into material properties [1]. In cuprate research, as a versatile probe of the electronic spin susceptibility, NMR was decisive for the field, early on (for reviews see [2–4]). However, the prevailing understanding of NMR was challenged, recently, as it was shown that the hitherto adopted magnetic hyperfine scenario appears to be incorrect [5], and very different conclusions about the magnetic response could be drawn [6,7].

Here, however, we will address NMR's capability to measure the charges at Cu and O atoms in the cuprate $CuO_2$ plane, where recent advances lead to surprising relations, as will be discussed here. While it is well known that the electric quadrupole interaction between nuclei and the local charges affect the NMR Zeeman splitting, quantitative conclusions can be rather complicated, if reliable first principle calculations are absent. In the cuprates, despite the rather ubiquitous $CuO_2$ plane, experiments showed, from the very beginning, that planar Cu and O can have vastly different splittings for different materials and that in most materials there are rather strong spatial inhomogeneities in terms of this local charge distribution. These observations were commonly attributed to effects from (inhomogeneous) charge reservoir layers that trigger the distribution of charge in the $CuO_2$ plane. The possibility to gain further insights was widely dismissed apart from few attempts, e.g., [8].

In recent years, our understanding of this electric hyperfine interaction has significantly improved so that we know, now, that NMR actually provides a quantitative measure of local charges at planar

Cu and O [9,10], i.e., one can determine the bonding orbital hole content of the Cu $3d_{x^2-y^2}$ ($n_d$) and the O $2p_\sigma$ ($n_p$) orbitals. As one would expect, for the parent compounds, the sum of $n_d + 2n_p = 1$, i.e., the hole contents $n_d$ and $n_p$ add up to the single parent hole nominally expected at planar Cu, cf. Figure 1. This simple result has unexpected consequences as one finds that the sharing of this single hole in terms of $n_d$ and $n_p$ varies substantially between the cuprates, and it turns out that the maximum $T_c$ ($T_{c,max}$) is proportional to the parent's O hole content $n_p$ [10,11], as indicated in Figure 2. Even the effect of chemical doping ($x$) can be quantified with NMR and the simple relation $1 + x = n_d + 2n_p$ is obtained, again to no surprise, but in terms of $n_p$ and $n_d$.

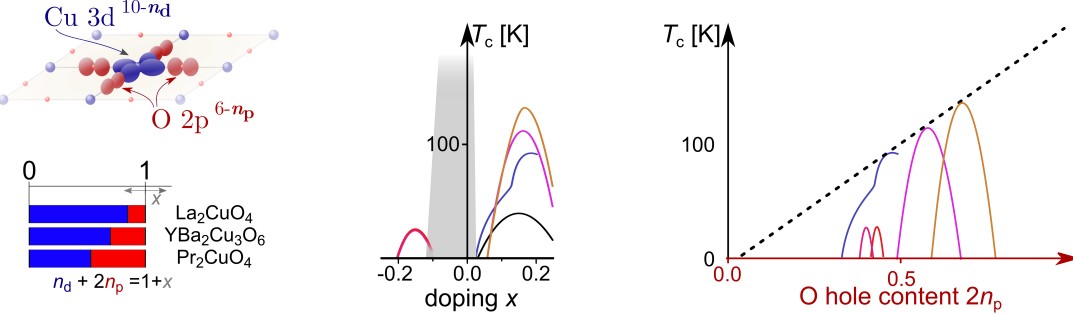

**Figure 1.** *Left, upper*: Sketch of the CuO$_2$ plane with hole contents $n_d$ and $n_p$ of the respective bonding orbitals Cu $3d_{x^2-y^2}$, and O $2p_\sigma$; NMR finds that $n_d + 2n_p = 1 + x$, but the hole sharing, i.e., the Cu-O bond covalency is material dependent (*lower*). *Middle*: Sketch of the commonly used phase diagram as generated by doping with electrons ($x < 0$) or holes ($x > 0$), with high-$T_c$ superconductivity as material-dependent dome-like regions (colored lines). *Right*: Considering $T_c(n_p)$ rather than $T_c(x)$ reveals scaling of the maximum $T_c$ with the O hole content $n_p$.

This new insight into the chemistry of the CuO$_2$ plane is addressed in this manuscript and consequences for relations to other cuprate properties will be discussed. These correlations between material chemistry and electronic properties may help theory as well as material scientists in the quest for room temperature superconductivity (that could occur for $n_p \approx 0.75$, as estimated from Figure 1).

In addition to measuring the average charge at planar Cu and O, NMR also provides a bulk histogram of the spatial charge density variation in the CuO$_2$ plane. Here, it turns out that almost all doped cuprates show pronounced planar charge variations independent of structural homogeneity. The only exceptions are a few very specific configurations of YBCO, but these show double peak patterns for planar O and a narrow Cu resonance, rather than broad distributions. For one such configuration, YBa$_2$Cu$_3$O$_{6.9}$, especially designed high pressure NMR experiments have recently identified the origin of this double peak pattern to be commensurate charge density variations on planar O that can order at elevated pressure [12]. Consequences of these findings will be discussed, as well.

## 2. NMR of Charges in the Cuprate Plane

Nuclei with spins $I > 1/2$ such as $^{17}$O ($I = 5/2$) and $^{63,65}$Cu ($I = 3/2$) possess a quadrupole moment that interacts with the electric field gradient at the nuclear site. This quadrupole interaction typically perturbs the Zeeman splitting (if the external magnetic field is strong enough), and information about the local charge symmetry can be conveniently obtained from the NMR spectra for various external field directions. Crystallographically inequivalent sites can be distinguished easily by their local charge symmetry, which can be used for a reliable assignment of NMR resonances to atoms in the unit cell. Therefore, such experiments were of great importance in early cuprate research. However, a deeper insight beyond spectral site assignment was limited. This was in part due to the fact that the rather universal CuO$_2$ plane showed widely varying quadrupole splittings, e.g., for planar $^{63}$Cu quadrupole frequencies were found to vary immensely from more than 35 MHz in La$_{2-x}$Sr$_x$CuO$_4$ [13] to near zero in Nd$_{2-x}$Ce$_x$CuO$_4$ [14]. In addition, first principle calculations appeared to give limited information [15,16], and it is not clear to what extent strong electronic

correlations restrict their reliability [17]. Aside from a few attempts at a quantification of planar charges [8], empirical models widely prevailed that overestimated the effects from the charge reservoir layers [18], even though this entirely failed to capture doped charges. From the early days of high-$T_c$ research, the important role of planar O was clear, since X-ray spectroscopy had unveiled in hole-doped cuprate perovskites the lack of the expected $Cu^{3+}$ configuration with one hole in the $3d_{x^2-y^2}$ orbital and a second hole in the Cu $3d_{3z^2-r^2}$ orbital, while on the contrary providing compelling evidence that the doping induces the insertion of holes in the O 2p orbitals [19] confirmed by electron energy-loss spectroscopy and many other methods.

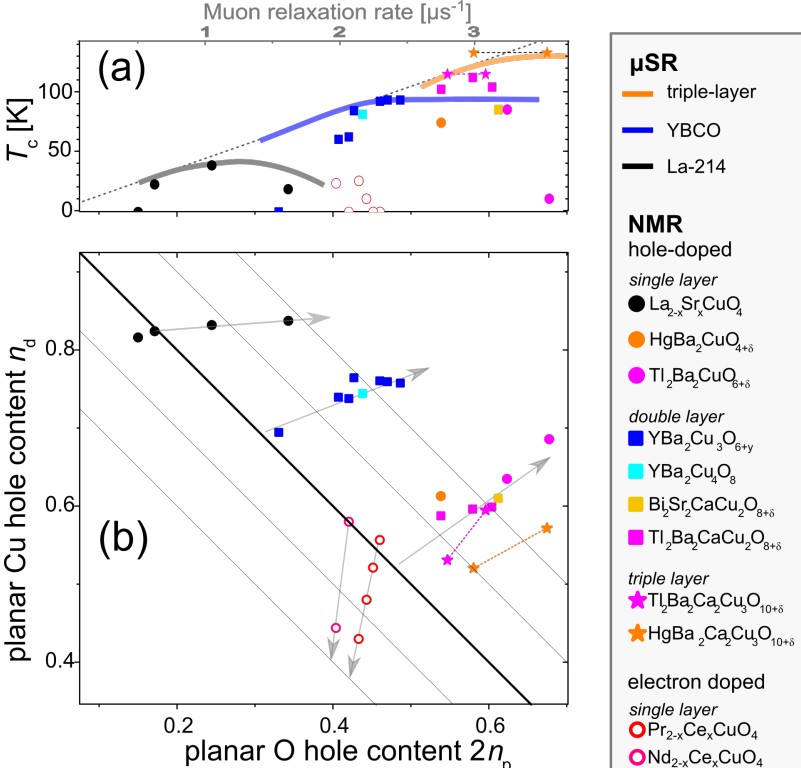

**Figure 2.** (**a**) $T_c$ of various cuprates (symbols) as function of planar O hole content $n_p$ [11]. Also shown, in thick lines, is the Uemura plot [20] with corresponding $\mu$SR rates as upper abscissa. The correspondence between $n_p$ determined from NMR, and superfluid density as follows from $\mu$SR reveals a material-chemistry origin for the ordering of different cuprates in the Uemura plot; (**b**) planar hole contents, $n_d$ vs. $2n_p$, for various cuprates; for triple-layer materials dotted lines connect data points for inner, less doped CuO$_2$ layers and outer, more doped layers [10]. The black diagonal, the 'parent line', marks the undoped state with one hole, i.e., $n_d + 2n_p = 1$; it separates upper-right hole-doped and lower-left electron-doped semi-planes. Grey lines indicate constant doping at 10% intervals. Grey arrows point towards higher, material specific doping, i.e., the distribution of doped charge; the arrows begin on the parent line, i.e., according to the sharing of inherent hole. The large differences in the planar hole distribution between Cu and O reveal differences in material chemistry that correlate with or even determine certain electronic properties, e.g., cf. (**a**).

In 2004, using electric hyperfine coefficients deduced directly from atomic spectroscopy, Haase et al. [9] showed that the average quadrupole splittings actually allowed for a precise quantification of holes doped into the CuO$_2$ plane, i.e., in terms of the changes in hole contents of the bonding orbitals Cu $3d_{x^2-y^2}$ ($n_d$) and O $2p_\sigma$ ($n_p$). A decade later, with new data available on electron doped materials, this analysis could be extended to the quantification of the total planar hole distribution as well [10]. Importantly, these new results lead to conclusions about the planar Cu-O-bond covalence, i.e., NMR was shown to quantitatively measure the sharing of the inherent, nominal Cu hole with O. For all materials where literature data are available, the hole contents $n_d$

and $n_p$ measured by NMR give in sum the planar hole content expected from chemistry. The naively expected relation $n_d + 2\,n_p = 1 + x$ is borne out for parent materials with just one inherent hole ($x = 0$), hole-doped cuprates ($x > 0$) as well as electron-doped cuprates ($x < 0$). It is important to stress that $n_d$ and $n_p$ represent the bonding orbital hole contents averaged at the NMR time scale, i.e., NMR measures localized as well as itinerant charges in these orbitals. While this method would be insensitive to, e.g., occupation of s-like orbitals, the quantitative agreement with material chemistry validates the approach.

An overview of the charge sharing is given in Figure 2b, which shows planar hole contents measured by NMR, $n_d$ vs. $2n_p$, for various cuprates. Undoped materials are found on the parent line (full black line) with $n_d + 2n_p - 1 = x = 0$, which separates the hole-doped half-plane (upper right) from the electron-doped half-plane (lower left). Unexpectedly, the sharing of the inherent hole between Cu and O differs markedly between different cuprate materials. The wide-spread materials along the parent line signals significantly disparate planar Cu-O bond chemistry. As we will discuss in the next section, this relates to substantial effects on the electronic properties. The different sharing of the inherent hole among the various materials must be due to a variations of planar bond covalency or, in a band picture, a material-specific charge transfer gap (CTG), cf. Figure 3. This is also consistent with the differing distribution of doped charges between Cu and O indicated by grey arrows in Figure 2b. For instance, holes doped in $La_{2-x}Sr_xCuO_4$ (black circles) almost exclusively go to planar O, since this material has the highest localization of the inherent hole on Cu and therefore the least covalent bond or largest charge transfer gap. On the other hand, the Hg, Bi and Tl based materials appear to have a comparatively small charge transfer gap, with $n_d \approx 2n_p \approx 0.5$ extrapolated for parent materials. Correspondingly, doped charges are shared equally such that $\Delta n_d / 2\Delta n_p \approx 1$.

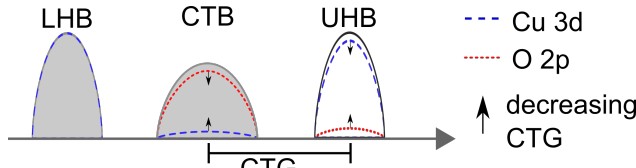

**Figure 3.** Sketch of undoped cuprate band structure. Upper (UHB) and lower (LHB) Hubbard band are dominantly of Cu 3d character, and the charge transfer band (CTB) of mainly O 2p. As the charge transfer gap (CTG) decreases, the O 2p contribution to the UHB and the Cu 3d contribution to the CTB increase (*arrows*).

The latter materials are somewhat different from other cuprate families in that these are doped by minute amounts of $O_\delta$ at interstitial sites and can be realized in different multi-layer configurations. Interestingly, the distribution of charge between $n_d$ and $n_p$ appears to be similar for all these materials independent of single-, double-, or triple-layer configuration. Note that, for triple-layer materials, the symbols (stars) connected by dashed lines represent less hole-doped inner $CuO_2$ layers and more doped outer layers.

We would like to stress that the material chemistry differences reflected in Figure 2b are measurable at room temperature and also in non-superconducting materials. As already mentioned, these differences also determine electronic properties like the maximum transition temperature ($T_{c,max}$) and the superfluid density ($\sigma_0$), cf. Figure 2a. It has therefore been proposed to consider the electronic cuprate phase diagram using planar hole contents $n_d$ and $n_p$ rather than the total doping [10,11]. In the following section, we highlight the advantages of this approach by considering electronic properties measured with other methods, where the material-specific differences appear to be linked to planar hole contents $n_d$ and $n_p$ measured by NMR. We note that trends we find in the material-specific planar charge distribution of hole-doped-cuprates, cf. upper right in Figure 2b, also show the expected correlation with apex-O distance to planar Cu [21] and the micro-strain [22] of the $CuO_2$ lattice.

## 3. Planar Charges and Electronic Properties

For our comparison of the material-specific planar charge distribution with electronic properties, we first consider the differences between cuprate families already evident in the spread of the undoped (parent) materials along the 'parent line' in Figure 2b, and continue to underdoped and optimally doped cuprates thereafter.

### 3.1. Parent Materials

The material-specific charge transfer gap reflected in the different sharing of charge, cf. Figure 2b, had previously been pointed out by computational methods [23]. Interestingly, Weber et al. [23] also concluded with the correlation between the maximum $T_c$ and the charge transfer gap (CTG). Using the calculated values of the charge transfer energy [23,24], we find the hole sharing of parent materials exhibiting the expected trend in Figure 4a. The ratio $2n_p/n_d$ decreases with increasing CTG, as we expect the inherent hole to localize on planar Cu, i.e., approaching $n_p = 0$ and $n_d = 1$. For decreasing CTG, $2n_p/n_d$ increases. Naively, one could expect equal occupation of Cu 3d and O 2p orbitals as the CTG vanishes, i.e., $n_p \approx n_d$, and $2n_p/n_d \approx 2$.

In parent materials, the large variation in hole sharing between different compounds should also be reflected in magnetic properties measured by other probes. Figure 4b shows the ordered magnetic moment on planar Cu measured by neutron scattering in the antiferromagnetically ordered state of cuprate parent materials dependent on the Cu hole content $n_d$ measured by NMR. The data show excellent agreement in the anticipated proportionality (grey line), the slope of which we find is 0.76 $\mu_B$ per Cu hole. Also relevant for neutron scattering is the fact that a varying, material-specific Cu-O bond covalency should affect the magnetic form factor. This is commonly associated with missing spectral weight in magnetic neutron scattering [25–27]. However, we are not aware of any study addressing the effect of a varying Cu-O hybridization on neutron scattering data over a wider range of different parent materials.

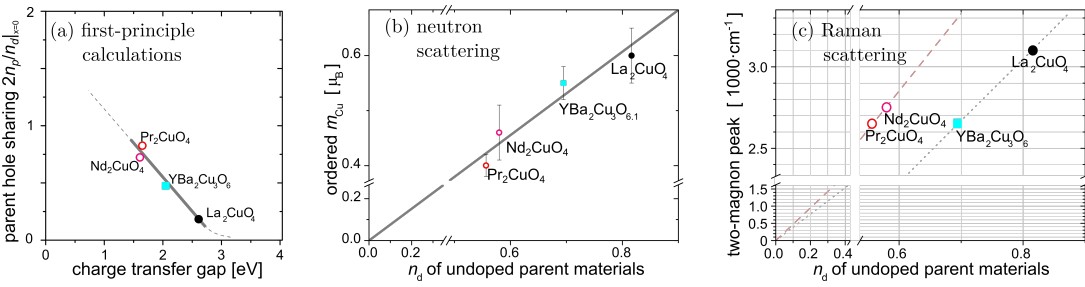

**Figure 4.** Parent charge transfer gap from NMR vs. other probes: (**a**) parent hole sharing ratio $2n_p/n_d$ from NMR vs. the calculated CTG [23,24] showing the expected trend, i.e., going to zero for increasing CTG ($n_p \rightarrow 0$) and towards two for vanishing CTG ($n_d = n_p$); (**b**) dependent on $n_d$ of the undoped parent materials, the ordered Cu moment in the AF state measured by neutron scattering [26] shows the expected proportionality (full line), the slope is 0.76 $\mu_B$/hole; (**c**) Raman two-magnon peak [28,29] vs. $n_d$ of undoped parent materials shows no clear overall behavior. Considering cuprates with (dotted line) and without (dashed line) apex O, separately, a proportionality might be present, but data are too limited for solid conclusions.

A varying CTG should also be detected by the two-magnon peak, $\omega_p$, in Raman scattering that measures the superexchange coupling. As shown in Figure 4c, it does not show an obvious dependence on the Cu hole content in parent materials. We note, however, that the width of the two-magnon peak is immense in comparison to the differences in $\omega_p$ between parent materials. In addition, hopping terms relevant for the superexchange energy are expected to be sensitive to the apex O distance and coordination [21], in which the materials in Figure 4c differ significantly. Considering only $Nd_2CuO_4$ and $Pr_2CuO_4$, both without apical O, a proportionality could be conceivable; this is less so for $La_2CuO_4$ and $YBa_2Cu_3O_6$, but these differ in apical O coordination and distance.

### 3.2. Underdoped Cuprates

The most prominent and still enigmatic feature in the underdoped regime is the pseudogap, manifestations of which are found below the pseudogap temperature $T^*$ that decreases with hole doping from several hundred Kelvin down to around $T_c$ near optimal doping. Owing to its gradual nature, the onset and amplitude of this phenomenon are not easily quantified. However, consensus seems to be that it is fairly universal [30], and we are not aware of reports concerning material-specificity of the pseudogap that could relate to the planar charge distribution.

Pseudogap features in NMR shift, where the phenomenon was first reported as a spin gap (the shift acquires a temperature-dependence above $T_c$, which is not expected for a Fermi liquid) [31], show clear material specificity in both planar $^{17}O$ and $^{63,65}Cu$. For the latter, scaling of the shift phenomenology with the charge distribution between Cu and O has already been identified [5]. However, as Haase et al. [5] have shown, the hyperfine scenarios thus far proposed are insufficient to describe the NMR shift data, and the solution of this issue promises further insight. In NMR spin-lattice relaxation of planar Cu, on the contrary, a review of all literature data showed no clear pseudogap manifestations at all. Rather, the Cu spin-lattice relaxation shows quite universal Fermi-liquid like behavior for all cuprates above $T_c$ [6].

Charge ordering phenomena in underdoped cuprates have been a focus of recent cuprate research [32], initially also triggered by NMR observations of charge ordering (CO) in underdoped $YBa_2Cu_3O_{6+y}$ at very high magnetic fields [33–35]. More recent NMR results indicate omnipresent charge density variations, i.e., for all materials and doping levels, which we address in the following Section 4. Over the last several years, an abundance of reports, predominantly using X-ray methods [32], have established CO within the pseudogap regime of most underdoped cuprates. The precise nature of CO and its role vis-à-vis superconductivity are the subject of current research and debate. While often suspected to be competing with superconductivity [36,37], recent studies on spatial inhomogeneity of CO using scanning micro X-ray diffraction have unveiled a quite complex quantum landscape made by the formation of nanoscale striped puddles in cuprates [38,39], which could favor superconductivity [40].

The onset temperature of CO, $T_{CO}$, in underdoped hole-doped cuprates, has a pronounced material dependence that appears to scale (to some extent) with the maximum $T_c$ at optimal doping; therefore, it also scales with $n_p$ determined from NMR (not shown). Interestingly, if we plot $T_{CO}$ vs. $n_d$, cf. Figure 5a, we find an overall trend of decreasing $T_{CO}$ with increasing $n_d$, also including electron-doped cuprates with very high CO onset temperatures [41]. Whether this has physical significance or is rather coincidental is not clear to us. However, we do note that, structurally, there is a trend of an increasing distance between planar Cu and apical O going from right to left in Figure 5a, note that $Nd_{2-x}Ce_xCuO_4$ does not have an apical O. In addition, we want to stress that the character of CO observed differs significantly between the various cuprates, e.g., in wave vector, doping dependence and other characteristics. The experimental separation of electronic charge density variation and strong spatial local lattice fluctuations associated with inhomogeneous dopant distribution [42,43], and their interplay [39] are increasingly investigated. Inspired also by the NMR results on differing CTGs, a recent study investigated the ground state of the three-band Hubbard model at 1/8 hole doping, and found a rich phenomenology of magnetic and charge ordering in dependence of the value of the CTG [44].

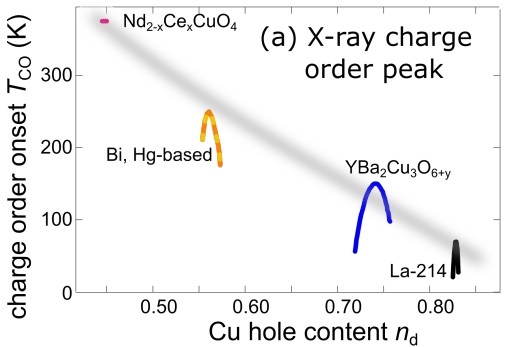
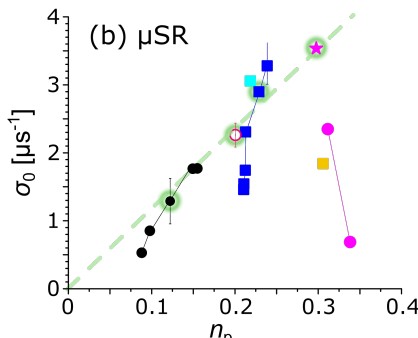

**Figure 5.** (**a**) charge order onset temperature for various cuprates [32,41] vs. corresponding Cu hole content $n_d$ from NMR; (**b**) $\mu$SR relaxation rate extrapolated to $T = 0$ K [20,45–47] for various cuprates vs. corresponding O hole content $n_p$ from NMR. Optimally doped materials (marked by green circles) appear to show proportionality (dashed line).

### 3.3. Optimal Doping

As already mentioned, the maximum $T_c$ ($T_{c,\max}$) scales with the planar O hole content of the parent materials, as shown in Figure 2a. Note that the increase in $T_{c,\max}$ with $n_p$ also holds if one considers electron-doped cuprates, separately, although with a diminished slope. Similarly, the trend holds when only looking at hole-doped single-layer cuprates, double-layer cuprates, or triple-layer materials. However, going from low to high O hole content in Figure 2a, it takes an increasing number of layers to unlock the maximum $T_c$ set by $n_p$.

Another important aspect of cuprate phenomenology that might be linked to the planar charge distribution is the curious, material-specific pressure dependence of the superconducting transition temperature. Generally, pressure appears to dope holes, i.e., $dT_c/dP$ is, respectively, positive, zero and negative for underdoped, optimal doped, and overdoped materials [48]. However, $T_c(P)$ of underdoped materials often exceeds the maximum values of $T_c(x)$ achieved with chemical doping, such that phenomenologically one distinguishes pressure-induced doping and inherent pressure contributions to $T_c$ [48], but also normal-state porperties as was found by, e.g., high-pressure NMR experiments [49]. In the context of planar charges shown in Figure 2b, this effect could be related to pressure-induced hole doping as well as charge redistribution between Cu and O, i.e., pressure leads to hole doping $2\Delta n_p(p) + \Delta n_d(p) = \Delta x(p) \geq 0$, but favoring $n_p$ over $n_d$ compared to chemical doping. Thereby, an increased O hole content ($n_p$) at optimal doping could lead to a higher $T_{c,\max}$ than chemically achievable. While literature data of $^{63,65}$Cu NMR under pressure are consistent with such a scenario, $^{17}$O pressure NMR data are needed for a verification (we are engaged in efforts to obtain such data).

In Figure 2a, we have also shown the famous Uemura plot [20] that revealed, for underdoped cuprates, a proportionality between $T_c$ and the superfluid density ($\sigma_0$) extrapolated to 0 K, ($\sigma_0$ as determined from the muon spin relaxation rate, $\mu$SR). The similarity of $\mu$SR and NMR data is quite surprising considering that $n_p$ reflects bonding chemistry measured at room temperature, while $\sigma_0$ is a (low energy) electronic property measured near $T = 0$ K. Thus, the charge sharing measured by NMR reveals the material chemistry origin of the ordering of La$_{2-x}$Sr$_x$CuO$_4$, YBa$_2$Cu$_3$O$_{6+y}$, and triple-layer cuprates in the Uemura plot. For a closer inspection, we plot in Figure 5b $\sigma_0$ vs. $n_p$ for various cuprates. At first glance, $\sigma_0$ does not reveal an obvious correlation with $n_p$. However, the superfluid density cannot be proportional to $n_p$, since the superfluid density must be nearly zero at the doping level where superconductivity emerges, and usually decreases in the overdoped regime. The O hole content, on the other hand, is non-zero already in the parent material and must increase and decrease monotonically with hole and electron doping, respectively. Consider just optimal doping for which data points are marked by green circles in Figure 5b. Whether achieved by electron or hole doping, we do find again a

proportionality between an electronic property and material chemistry, the physical significance of which is yet to be understood.

In summary, the significant material chemistry differences of planar charges as revealed by NMR show unsuspected scaling with a number of electronic properties over all cuprate families. With our limited expertise in other probes and a limited number of reports comparing different cuprate families quantitatively, there is likely much more to be learned when considering the complex cuprate phenomenology in terms of the planar hole contents $n_{\mathrm{d}}$ and $n_{\mathrm{p}}$, rather than total doping $x$.

## 4. Planar Charge Density Variations

Since NMR measures charges with atomic resolution, one may expect it also to be sensitive to charge density variations or even charge ordering phenomena, as well. The average quadrupole frequencies of planar $^{63,65}$Cu and $^{17}$O provide the quantitative measure of the average bonding orbital hole contents, and a spatial distribution of charge must result in distributions of quadrupolar frequencies, which are directly determining the satellite lineshapes for both nuclei.

Early on, it was observed that most cuprate materials appeared to be very inhomogeneous, in the sense that excessive quadrupolar broadening of the satellite spectra was found. This was generally deemed to be of lesser interest and attributed to structural inhomogeneity and limited sample quality of these doped materials. In particular, since certain configurations of the YBCO-family of materials exhibit no excessive quadrupolar linewidths, it was concluded that charge inhomogeneity is not relevant for cuprate physics (a different conclusion is reached below). For all other cuprate materials, including other YBCO-materials, significant charge inhomogeneity was found, also in high quality single crystal samples. Even, e.g., for single-layer $HgBa_2CuO_{4+\delta}$, considered to have a highly homogeneous $CuO_2$ plane, i.e., without buckling or structural distortions and the $O_\delta$ dopants far away from the planes, Cu and O NMR still show large quadrupolar linewidths that cannot be accounted for structurally [50]. On the other hand, even in apparently highly inhomogeneous systems like electron-doped $Pr_{2-x}Ce_xCuO_4$ with an astonishing small average $^{63}$Cu quadrupole frequency ($\leq 1$ MHz), one finds immensely broadened ($\approx 7$ MHz) satellite spectra that precisely follow the local symmetry of the bonding orbitals, i.e., it shows no signs of structural distortions but rather a simple variation of hole content [51].

The only materials showing no significant quadrupolar broadening are still those few found early on, namely: $YBa_2Cu_4O_8$ and some specific, well-ordered phases of $YBa_2Cu_3O_{6+y}$, however, these very materials all exhibit double-peak spectra for their planar O satellites. Complicating a simple understanding, these materials are slightly orthorhombic, i.e., on symmetry grounds there are slightly different Cu-O-Cu bond lengths, which could explain the spectra (or in the case of de-twinned ortho-II and ortho-VIII phases these double lines could signal empty and full adjacent chain sites [35]). However, there are some inconsistencies if one tries ascribing the double peak features to the orthorhombicity, e.g., the peak splitting does not follow changes in orthorhombicity [52], while the size of the splitting is similar to the linewidths observed in all the other systems.

Already some time ago [52,53], it was shown that an alternate explanation is possible: Since NMR probes the bulk of the material giving a histogram of the sites of the nucleus investigated, commensurability is central to the effect any variation would produce spectrally. If a charge density variation has a certain commensurability with the lattice, it could produce as little as two different planar oxygen charges in the plane consistent with the double-peak spectra seen for certain YBCO-materials. An overall incommensurability, of an otherwise similar charge density variation, in other systems would create broad lines of similar extent, as observed for other configurations of YBCO. Unfortunately, the differences between the two scenarios, structural vs. electronic origin of the double-peaks, are difficult to prove by experiment.

Only very recently, the structural interpretation was disproved using high pressure NMR on optimally doped $YBa_2Cu_3O_{6+y}$ [12], cf. Figure 6. Reichardt et al. [12] investigated the charge symmetry at planar O and Cu at moderate pressures and temperatures that do not affect the chemical structure at

all. Nevertheless, they observed significant changes of the local charge symmetry proving its electronic origin. The Cu and O data revealed unordered, short-range domains of a d-wave charge density wave on O with an amplitude of $\delta n_p \approx \pm 1\%$. At elevated pressure, the amplitude increases slightly and the domains align and possibly grow, such that at 18 kbar and 100 K one has aligned O charge order throughout the bulk in a nonetheless twinned crystal. As a local probe, NMR cannot easily determine the wavelength of the variation; however, with a bulk probe's confidence NMR can see if all local patterns are aligned.

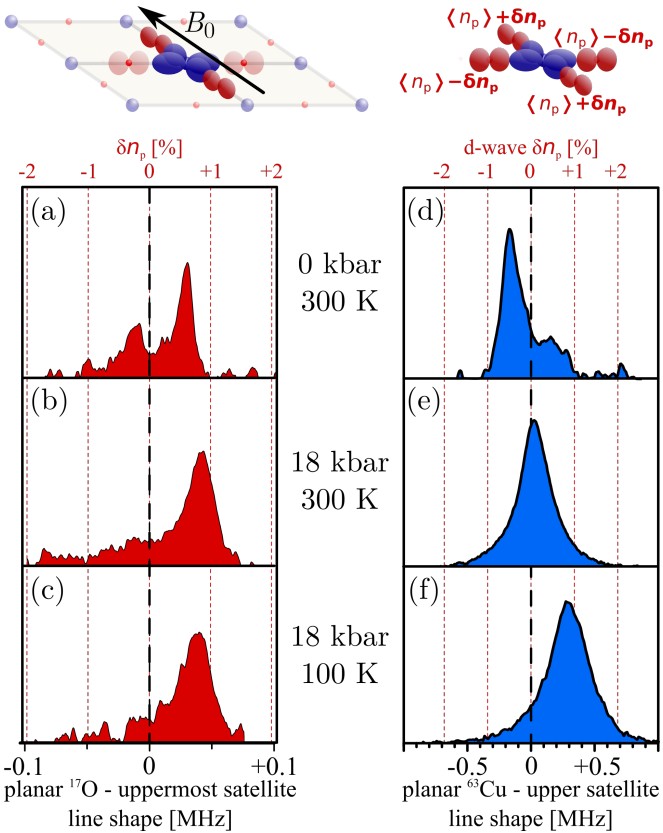

**Figure 6.** Oxygen charge ordering scenario in $YBa_2Cu_3O_{6.9}$ determined by high pressure NMR from (**a**–**c**) planar $^{17}$O, and (**d**–**f**) planar $^{63}$Cu uppermost satellites with the external magnetic field $B_0$ applied along an in-plane bond direction (*sketch on top left*) of a twinned crystal. At ambient conditions (**a**,**d**), both nuclei show a double peak centered around the expected average hole contents (dashed vertical line). With increasing pressure (**b**), one O peak diminishes and (**e**) the Cu line loses asymmetry. Upon cooling (**c**,**f**), both nuclei show a single peak, away from what is expected from average hole contents. Structural explanations based on orthorhombicity fail because of, e.g., unchanged twinning throughout, reversibility in $T$ and $P$ of the observed effects, apparent intermediate loss of orthorhombicity (**e**) and various other features, observed by Reichardt et al. [12]. For an O charge density variation of a given amplitude the so-called d-wave arrangement of charge (*sketch on top right*) causes the maximum effect on the asymmetry of the Cu electric field gradient. It changes the Cu quadrupole splitting as indicated by the upper abscissa and the dotted lines for (**d**–**f**). The disappearance of the O resonance from O with the smaller hole content, $\langle n_p \rangle - \delta n_p$, in (**c**) concomitant with the presence of a larger signal from O with $\langle n_p \rangle + \delta n_p$ means global O d-wave ordering, for the bond axis along $B_0$. The Cu asymmetry in (**f**) also points to global d-wave ordering, even of the same amplitude. This proves the electronic, not the structural origin of the double peak satellites. Being at ambient conditions (**a**,**d**) must be due to unordered, likely short-range O d-wave variation. Although not their origin, the crystals orthorhombicity may still affect the direction of the d-wave domains' wavevectors. Unordered short-range O d-wave charge variations at ambient conditions (**a**,**d**) grow in amplitude, nearly fully aligning at elevated pressure and decreased temperature (**c**,**f**).

Having established an electronic charge variation as the origin for the double peak feature in $YBa_2Cu_3O_{6.9}$, a reconsideration of the formerly dismissed charge inhomogeneity in all other cuprates appears expedient. Figure 7 shows $^{17}O$ satellite transitions for various cuprate materials. Comparison with the indicated magnetic linewidths (⊢⊣) reveals significant O charge variations ($\delta n_p$) in all cuprates of somewhat comparable amplitude. Interestingly, some underdoped single crystals appear to also show double peak features despite a tetragonal unit cell, e.g., $HgBa_2CuO_{4+\delta}$ and $Pr_{1.9}Ce_{0.1}CuO_4$.

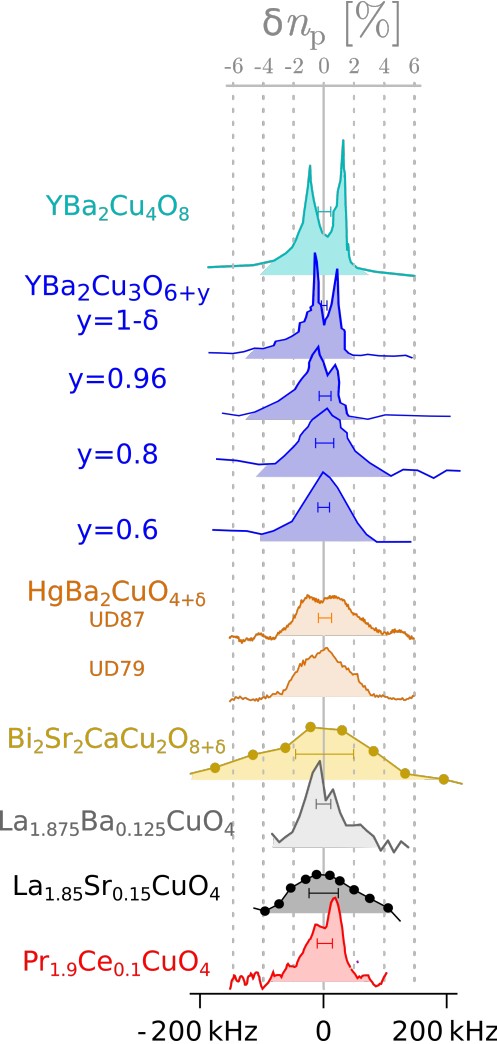

**Figure 7.** Uppermost satellites of planar $^{17}O$ with $c \parallel B_0$ as observed in various cuprates [12]. A significant O charge variation is evidenced for most materials by broadening well beyond magnetic linewidths, which is indicated by ⊢⊣; the upper abscissa indicates the necessary charge variation in terms of $\delta n_p$ of O hole content about average value $\langle n_p \rangle$. Certain YBCO materials show narrow, double peak satellites, which were usually explained with the unit cell's orthorhombicity, but for $YBa_2Cu_3O_{6.95}$ they were now shown to be due to commensurate charge density variations on O [12]. In addition, even some materials with tetragonal unit cells (e.g., the Hg and Pr based materials) can show double peak features where a structural explanation cannot be given. Sufficiently coherent charge density variations can lead to special NMR lineshapes as the nuclei sample the local charge.

Similar to planar $^{17}O$ NMR, NMR of planar $^{63,65}Cu$ reveals significant charge inhomogeneity for almost all doped curates [54]. Quantitatively, one finds variations of bonding orbital hole contents from around 1% to beyond 5% for both $n_p$ of O and $n_d$ of Cu. As indicated above, while NMR provides a quantitative measure of charge, the lineshapes revealing charge variations always only represent a bulk histogram for a local pattern. For instance, symmetry and wave vector of a possible

charge density wave are not directly accessible. A commensurate charge density wave could give characteristic spectral features like a double peak [52,53], but incommensurability, limited correlation length, or incoherent domains would result in featureless broadening. The high-pressure NMR results of Reichardt et al. [54] represent a particularly fortunate finding as spectral features of both Cu and O NMR independently prove a d-wave charge variation of $n_p$ of the same amplitude, and this charge variation orders at elevated pressure throughout the bulk of a twinned crystal. A vital question with regard to this bulk charge ordering is what breaks the in-plane symmetry, i.e., why does the charge ordering seen in Figure 6c,f align (reversibly in *T*) in this way as opposed to the 90° orthogonal in-plane bond direction? The magnetic field along this direction of the twinned crystal appears to be the only viable candidate; however, it is unclear how specifically the magnetic field affects the the O d-wave charge density variation. We note that already previously high [33–35] and even moderate [54] magnetic fields have been found to affect charge variations seen by NMR.

The omnipresent charge density variations indicated by NMR, which are found to be largely independent of doping and temperature, appear to be at odds with what was reported with other probes, e.g., charge ordering phenomena in the underdoped regime below an onset temperature ($T_{CO}$). However, e.g., $T_{CO}$ as determined in X-ray experiments marks the decrease of the charge ordering peak height below measurability and is accompanied by a broadening that signifies a decreasing correlation length. Charge density waves of short correlation lengths can therefore not be excluded, and these would in terms of NMR lead to featureless broadening as it is generally observed.

## 5. Conclusions

In addition to probing magnetic properties in the bulk, NMR of planar $^{63}$Cu and $^{17}$O also provides a quantitative measure of the local distribution of doped and inherent charges in the $CuO_2$ plane. Thereby, it has revealed significant differences in planar bonding between different cuprate families. These material chemistry differences reflect differing charge transfer gaps or a varying Cu-O bond covalency. These appear to determine many electronic properties observed by various probes, e.g., the maximum $T_c$ and the superfluid density at optimal doping, but also magnetic properties in undoped or underdoped cuprates. The relevance of the variable hole sharing, i.e., band contributions of O $2p_\sigma$ and Cu $3d_{x^2-y^2}$, should inform theorists with regard to the mechanism of cuprate superconductivity as well as chemists regarding possible new high-$T_c$ materials. Measuring local charges quantitatively, NMR also measures their variation. Charge density variations on Cu ($n_d$) as well as O ($n_p$) with amplitudes of 1–5% appear to be omnipresent in doped cuprates. Although the symmetry of charge variation is inherently difficult to characterize with NMR, O charge density variations in optimally doped $YBa_2Cu_3O_{6+y}$ could recently be shown to have d-wave character using high-pressure NMR.

**Author Contributions:** All authors contributed equally to the conceptualization; the original draft was prepared by M.J.; review and editing was performed by M.J., A.E. and J.H.; supervision by J.H., as well as funding acquisition.

**Funding:** This research was funded by the University of Leipzig, the Free State of Saxony, and the Deutsche Forschungsgemeinschaft (DFG) grant number HA 1893/18.

**Acknowledgments:** We thank Antonio Bianconi for helpful discussions.

**Conflicts of Interest:** The authors declare no conflict of interest.

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
