# Peer review of "Tc and Other Cuprate Properties in Relation to Planar Charges as Measured by NMR"

_condensedmatter, doi:10.3390/condmat4030067_

Round 1

Reviewer 1 Report

The paper is well written and provides an excellent review on the authors contributions and other researchers results on NMR data for cuprates. The conclusions about the role played by holes on Cu, respectively oxygen, are interesting and inspiring for the theoretical understanding of superconductivity in cuprates.

Author Response

We thank the Reviewer for the positive comments.

Reviewer 2 Report

The authors present the research results in accordance with the title "Tc and other cuprate properties in relation to planar charges as measured by NMR".

The results provide an overall review of the research done by (mostly) the authors themselves over the period of more than a decade. While it can be easily seen that studying this topic has been an exhausting task, I don't it is now the time to lower standards. In the manuscript, there is some new information provided, but most of the work is a summary of previously published work. MDPI has a rule regarding publication ethics: “Manuscripts should only report results that have not been submitted or published before, even in part.”, which the submitted manuscript unfortunately breaks. I have no problem when authors use their previous work to build on a topic, but it is difficult to find something new in this manuscript. If the authors can provide additional data on each of the reported and previously published results, that would further strengthen their results I would be more inclined to looking at it as a new piece of work.

For instance:

(i) can the authors provide a full angle dependence of the 17O quadrupolar splitting presented in ref. 12, to show that a sample misalignment and/or local misalignment of the EFG tensor is not the cause of a mismatch in Vzz/2 values and Vxx, Vyy? Is there an xray or neutron study that shows there is no tilting of the CuO4 plaquette with pressure? An earlier study (R. Ofer and A. Keren, Phys. Rev. B 80, 224521 (2009)) found that there is a tilting of the EFG principle axis in the YBCO – is there a way to show that this does not happen in other cases of cuprates and influence the provided explanation? 

(ii) In lines 184 – 189, there is a statement “Similarly, the trend holds when only looking at hole-doped single-layer cuprates, double-layer cuprates, or triple-layer materials.” Is it possible to provide some data on the quadrupolar splitting of 17O for e.g. a four-layer or four-layer cuprate, to show this deviation? Is there an explanation on why this scaling breaks down? The cuprate and NMR community would be interested in such data.

As I said earlier, I have no problem when authors use their previous work to build on a topic, but what is new here? 

The authors need to properly reference fig.2, as it is already published in ref. 11.

For all the figures that appear in other publications - have copyrights and permissions from the original journal been checked before?

Author Response

Reply to Reviewer 2:

Concerning the alleged lack originality of the work, it is true that we do not present new experimental data (we neither state nor even imply this anywhere), but we use existing data and compare them with data from other methods. Using Figures 1 and 2 we recapitulate what NMR of planar charges has previously revealed about varying material chemistry and how this sets the maximum Tc, with the new sketch Fig. 3 added for illustration. The following five plots in Figs. 4 (a-c) and 5 (a,b) reveal entirely new, unpublished correlations between charges measured by NMR and material-dependent electronic properties determined with other methods. We believe that each of these would even warrant publication by itself, as each appears to reveal the material-dependence of an electronic property to be originating in planar band covalency/charge transfer gap.

While 4(a) & (b), and possibly even (c), could have been expected, it is nonetheless worth giving experimental proof. On the other hand, explanations for the trend in TCOin 5(a) and the proportionality in 5 (b) just for optimal doping, are not obvious. In our view all of these unpublished relations of electronic properties should definitely be of interest, particularly to theorists, as other Reviewers agree.

Reviewer 2 specific comment: (i) can the authors provide a full angle dependence of the 17O quadrupolar splitting presented in ref. 12, to show that a sample misalignment and/or local misalignment of the EFG tensor is not the cause of a mismatch in Vzz/2 values and Vxx, Vyy? Is there an xray or neutron study that shows there is no tilting of the CuO4 plaquette with pressure? An earlier study (R. Ofer and A. Keren, Phys. Rev. B 80, 224521 (2009)) found that there is a tilting of the EFG principle axis in the YBCO – is there a way to show that this does not happen in other cases of cuprates and influence the provided explanation?

Reply to Reviewer 2 specific questions:Concerning (i), the Supplementary Figure 9 of ref 12 shows the alignment of the crystal within the pressure cell using NMR of both 63Cu and 17O. Additionally, there are various spectroscopic features that explicitly rule out misalignment, e.g., in Fig. 2 of ref 12 misalignment of B0 from the in-plane bond direction could only cause a diminished splitting in Fig. 2(f), but not the observed increase. Also, while that figure only shows the uppermost satellite, all transitions where measured, and can only be explained consistently including higher order quadrupole shifts with Vyy aligning in this direction. 

We are not aware of x-ray or neutron studies indicating tilting of the CuO4 plaquettes. The study of Ofer et al. seems to have been done on aligned powders, but details on the samples are sparse and we could not find any data (XRD-rocking curves) on how well these are aligned, but one expects on general grounds a large spread of angles. In another study (Jurkutat, M., Haase, J., & Erb, A. (2013) J SUPERCOND NOV MAGN 26(8), 2685–2688) we addressed this for a single crystal of Pr1.85Ce0.15CuO4. Here, Vzz on average is close to zero (≤1MHz) but broadly distributed (≥5MHz), nonetheless all spectral intensity collapses to a narrow spectral region when the crystal is oriented in the magic angle (54.7° between B0 and c) showing that the EFG variation is only scalar and not in terms of orientation of Vzz.

Reviewer 2 specific comment:(ii) In lines 184 – 189, there is a statement “Similarly, the trend holds when only looking at hole-doped single-layer cuprates, double-layer cuprates, or triple-layer materials.” Is it possible to provide some data on the quadrupolar splitting of 17O for e.g. a four-layer or four-layer cuprate, to show this deviation? Is there an explanation on why this scaling breaks down? The cuprate and NMR community would be interested in such data.

Reply to the Reviewers 2 specific question: Concerning (ii), we are not aware of 17O quadrupole data on four- or five-layer materials, and we would need data on different materials at optimal doping to judge whether the trend also holds here or breaks down. 

Reviewer 2 specific comment: As I said earlier, I have no problem when authors use their previous work to build on a topic, but what is new here? 

The authors need to properly reference fig.2, as it is already published in ref. 11. For all the figures that appear in other publications - have copyrights and permissions from the original journal been checked before?

Reply to Reviewer 2 specific comment:We have added the requested reference.

Reviewer 3 Report

The authors present a nice overview on the local charge configuration probed by NMR in the cuprates. First of all they point out that in studying the correlation between Tc and charge doping one should carefully look not only at the total charge content but also at the charge unbalance between copper 3dx^2-y^2 and oxygen 2p_x,y orbitals, which is determined by the charge transfer gap. In particular, they point out that the maximum Tc of a given family of cuprates is proportional to the O 2p orbital hole content in the parent compounds. Then, they discuss the presence of a charge modulation which is ubiquitous in the cuprates, causing a broadening of the quadrupolar frequencies both of 17O and of 63,65Cu along with the results by other techniques, mainly X-ray scattering, as well as the effect of high pressure on the NMR spectra providing further support to the evidence of an intrinsic charge modulation. The subject is quite timely and this overview and the points raised are quite interesting and deserve to be published. My only concern is about the text readability in some parts, in particular in the abstract and introduction. Moreover, in several parts the sentence starts with “And” which is not gramatically correct. With a revision of the English throughout the text and an improvement of the readability the manuscript can be suitable for publication.  

Author Response

Thank you, we have implemented the requested changes.

Reviewer 4 Report

Authors highlight the significance of NMR to predict some of the electronic properties of CuO2 based superconductors. The main focus is the capability of NMR to measure the charges at Cu and O atoms in the cuprate CuO2 plane. The quantitative measure of local charges at planar Cu and O  and hence the bonding orbital hole content can give new insight in the field of CuO2 based superconductor.

Authors proposed to consider the electronic cuprate phase diagram using planar hole contents nd and np rather than the total doping and tries to correlate different electronic properties measured with other methods.

I agree with the authors that the material chemistry differences linking to electronic properties might help towards finding new high Tc materials.

Author also claims that NMR is sensitive to charge density variations in both planar bonding orbitals. Regarding this, I would like authors to further explain the occurrence of two peaks, other than broadening, in Fig 6 and Fig. 7. 

Author Response

The Reviewer wants us to elaborate on two-peaks vs broad-line satellite line shapes. We have therefore reformulated and extended our paragraph addressing this issue (lines 266-274).

The present manuscript is meant to show a broader audience what local charges in the CuO2 plane measured by NMR reveal about cuprate superconductors. We have therefore deliberately refrained from NMR technicalities and spectroscopic details to keep a broad appeal. and we hope our changes meet the Reviewers expectation.

Round 2

Reviewer 2 Report

The authors have replied some of my questions and comments, but some of them remained open. The general approach to their answers is "we don't really care", which is not the way to go about it. It is particularly worrisome that their response, as researchers, to the question of the originality of their work is "..it is true that we do not present new experimental data (we neither state nor even imply this anywhere)".

As I mentioned in the first review, MDPI has a rule regarding publication ethics: “Manuscripts should only report results that have not been submitted or published before, even in part.”. Therefore, by submitting to MDPI, you agree to these terms and conditions.

Now, let's turn the focus on the physics.

The Figures 4 (a-c) and 5 (a,b) really do show new, unpublished correlations between charges measured by NMR and material-dependent electronic properties. The authors make use of already published data and show correlations that is consistent with their interpretation.

However, my issue was in any case with the data shown in Figs. 2, 6, and 7, for which I asked a bit more elaboration, so that it is not a duplicate of their earlier work.

I asked for a full angle dependence of the 17O quadrupolar splitting presented in ref. 12 (and in Figs. 6 and 7 in the present manuscript), to show that a sample misalignment and/or local misalignment of the EFG tensor is not the cause of a mismatch in Vzz/2 values and Vxx, Vyy. This seems to me a reasonable request as the interpretation of data relies on the value of the quadrupolar splitting at the planar Cu and O sites, and such an angle dependence would resolve any existing doubts.

The Supplementary Figure 9 of ref 12 that the authors refer to shows a partial angle dependence of the 17O quadrupolar splitting within the CuO2 planes. For 63Cu central line position it shows raw data in which the magnetic shift has a contribution from the shift of the large quadrupolar splitting, and thus needs to be calculated.

While the former is asking just for an expanded graph, the latter type of data are rather easy to extract by a simple exact diagonalisation of the NMR Hamiltonian. At the same time, these data are quite important for the convincing the community. Therefore, I still request that the authors provide additional graphs on this point prior to publication.

Author Response

Response to Reviewer 2

In the following we repeat the review (in italics) with our responses interspersed at the appropriate points.

Reviewer 2:

The authors have replied some of my questions and comments, but some of them remained open. The general approach to their answers is "we don't really care", which is not the way to go about it. It is particularly worrisome that their response, as researchers, to the question of the originality of their work is "..it is true that we do not present new experimental data (we neither state nor even imply this anywhere)".

As I mentioned in the first review, MDPI has a rule regarding publication ethics: “Manuscripts should only report results that have not been submitted or published before, even in part.”. Therefore, by submitting to MDPI, you agree to these terms and conditions.

Now, let's turn the focus on the physics. The Figures 4 (a-c) and 5 (a,b) really do show new, unpublished correlations between charges measured by NMR and material-dependent electronic properties. The authors make use of already published data and show correlations that is consistent with their interpretation.

Our Response:

We are very sorry that Reviewer 2 got the impression we would not care about this issue. We tried to address every point as directly as possible. As the reviewer has correctly pointed out, our manuscript reviews previously published data and deduces new insights from them, i.e. from local charges in the CuO2 plane determined with NMR. As a result, “new, unpublished correlations between charges measured by NMR and material-dependent electronic properties” are reported. We only wanted to emphasize that we agree that the data our new results are based on, are not new.

Reviewer 2:

However, my issue was in any case with the data shown in Figs. 2, 6, and 7, for which I asked a bit more elaboration, so that it is not a duplicate of their earlier work.

Our Response:

We have made corresponding changes to the manuscript, i.e., we added explanations in the captions of those figures; we did not dare to change the figures themselves, so that the Editor does not need to seek confirmation from the other 3 Reviewers.

Reviewer 2:

I asked for a full angle dependence of the 17O quadrupolar splitting presented in ref. 12 (and in Figs. 6 and 7 in the present manuscript), to show that a sample misalignment and/or local misalignment of the EFG tensor is not the cause of a mismatch in Vzz/2 values and Vxx, Vyy. This seems to me a reasonable request as the interpretation of data relies on the value of the quadrupolar splitting at the planar Cu and O sites, and such an angle dependence would resolve any existing doubts.

The Supplementary Figure 9 of ref 12 that the authors refer to shows a partial angle dependence of the 17O quadrupolar splitting within the CuO2 planes. For 63Cu central line position it shows raw data in which the magnetic shift has a contribution from the shift of the large quadrupolar splitting, and thus needs to be calculated.

While the former is asking just for an expanded graph, the latter type of data are rather easy to extract by a simple exact diagonalisation of the NMR Hamiltonian. At the same time, these data are quite important for the convincing the community. Therefore, I still request that the authors provide additional graphs on this point prior to publication.

Our Response:

In order to comprehensively reply and hopefully disperse any doubts about the validity of the conclusions drawn in reference 12, we have found it helpful to structure our response into (I) why 17O angular dependence beyond the published data is not feasible, (II) why crystal misalignment can nonetheless be excluded, and (III) a discussion of the proposed local misalignment in terms of how this would affect the spectra and how this compares to the data.

(I) We completely understand the reviewer’s argument that the knowledge of the full angular dependence of the splitting of planar O could easily disperse any doubts concerning a misalignment. Upon closer inspection, however, it is for a number of reasons neither feasible nor even possible to get the desired information in this case:

-First of all, we are somewhat limited in terms of orientation given that the micro-crystal fixed inside the micro-coil of an anvil cell device. So measurements along the specific crystal axis that is aligned with the coil axis are impossible (B0 || B1). Also for any coil orientation away from perpendicular B0 our excitation conditions worsen and of course signal is diminished. In order to avoid this and achieve the greatest possible access to any desirable crystal axis, a twinned crystal was placed such that an in-plane bond axis (a/b) is along the coil axis, such that we can measure along c, the other in-plane bond-axis (b/a) and anywhere in between. This was fairly successful as the small correction angles determined in Suppl. Fig. 9 show.

-More importantly, the full angular dependence quadrupole frequency of planar O is not measurable with sufficient spectral resolution for directions away from the crystal’s symmetry axes. Given the number of O sites, with more or less asymmetric EFG tensors, the fact that we are measuring a twinned crystal and 17O having I=5/2, we can expect in dependence on the angle up to seven quintuplets of resonances . For most of these the angular-dependent splitting will go through 0 rotating from c-direction into the plane. Given the linewidths we have, the abundance of resonances, the diminished splittings at intermediate angles and the weak signal from our micro-crystal, a full angular dependence is unfortunately not feasible and also would not have the sufficient spectral resolution to provide the desired data.

-The authors MJ and JH have the experience that even in big high-quality single crystals of cuprate materials with simpler orthorhombic unit cells and much less O sites, the precision required to attain the desired information at intermediate angles would still be hard to achieve due to linewidths (we have performed such measurements on electron-doped systems before where there was a favorable resolution).

-It was therefore decided to determine the c-axis direction using the 63Cu central line (Supp. Fig.9(a)) and, once the CuO2-plane (perpendicular to c) was determined, to use planar O splitting to align the twinned crystal such that B0 is along an in-plane bond direction. Along the bond O splitting has a global maximum and interference with resonances of other sites does not present a problem here.

(II) Nonetheless, in Reichardt et al. ample evidence is presented that excludes misalignment of crystal with respect to B0. For instance:

-The Cu quadrupole satellites measured with cIIB reproduce the quadrupole frequency and linewidth determined with NQR for every pressure and temperature, such that the orientation cIIB should be beyond doubt throughout all conditions. Therefore we can clearly exclude significant angular offset from c\perp B0 for our measurements in along the bond (abIIB0).

-For both Cu and O along the bond at 18kBar and 100K, we find a splitting bigger than what is expected from the other crystal directions. Any misalignment away from the bond direction could only result in a decrease of the measured splitting.

-The reversible temperature-dependent effects, e.g., in  Fig. 3 of Reichardt et al. are only explicable by a change in EFG (charge) symmetry. In particular, crystal misalignment, again here, could only lead to a diminished splitting, not the measured increase.

-The only structural explanation for the symmetry changes measured that we could find would be a curiously reversible detwinning/twinning when going from 300K down to 100K and back. This can be excluded for various reasons, cf. Methods section of ref.12.

-Additionally, owing to the traceless EFG an in-plane misalignment of B0 with the bond direction could not explain the observed difference in Fig 5c of ref12 between expected(red dashed line) and measured (solid blue line) first moment of the outermost satellites. The EFG is traceless for any three perpendicular directions, not just the principal axes. While we are dealing with EFG distributions rather than precise values, we could still have curious linewidth effects (2nd and moments) but not in the center of gravity. The only explanation we found is that the EFG distribution we measure along the bond (Fig.5c) and the one in-plane perpendicular to the bond (Fig.4b top panel) are from two EFG-wise different subsets of planar O nuclei. In cIIB we measure both together. Concluding from the combined set (in c direction) and one subset (measured perp. to the bond) on the other subset (along the bond) then results in the discrepancy.

(III) Concerning the proposed scenario of local misalignment, we have frankly thus far not considered this. But one should of course discuss possible effects and compare them against experiment, assuming an angular distribution CuO4-symmetry axes about the crystal c-axis:

-First for Cu, the greatest effect should be in terms of angular dependent 1st-order quadrupole effects, as these far exceed higher-order quadrupole effects as well as magnetic shift.

*Here the comparison between NQR and cIIB in suppl. Fig.3 is particularly useful to check. While the NQR frequency is independent on angular distribution, for NMR a distribution about the c-axis would result in a low frequency tail, the shape and extent of which would reflect the distribution. While we generally do not observe pronounced low frequency tails, there appears to be a slight discrepancy on the low-frequency side in Suppl. Fig.3(f). Note, however, that a relevant angular offset of say 10° would give a shift of around 1.4MHz. The observed discrepancy of around 100 kHz could signify a deviation of less than 3°.

*Similarly comparing B perpendicular to c with NQR data, cf. Suppl. Fig. 4, an angular distribution would cause pronounced low-frequency tails for the NMR satellites shown. As for the c-direction, the in-plane Cu satellites show no relevant deviation.

*We cannot exclude very small variations, the evidence does certainly not imply relevant effects.

-For planar O also the greatest effect should be in terms of angular dependent 1st-order quadrupole, i.e. broadening around the principle axis. Effects would, however, be somewhat blurred due to the double peak pattern and further by the lack of asymmetry of the site.

*Generally, for the assumed axial distribution about the c-axis one expects a low-frequency tail for the quadrupole splitting along the bond ( for angles deviating from Vzz) as well as for in plane perpendicular (for angles deviating from Vyy), both represent global extrema for the direction dependent values Vij of the EFG. For cIIB (Vxx) the measured splitting would also decrease for the field shifting to the Vzz direction, but increase towards Vyy. Therefore, a local angular distribution would result in a low-frequency as well as high-frequency tail on the splitting in cIIB, but with the latter effect less pronounced.

*Comparing these expected effects with the data, we find for the bond direction that tails towards lower splitting could be interpreted, cf. Fig.5. On the other hand, the lineshapes might as well be made up of two lines of different shape, e.g., suppl. Fig.6. For B perpendicular to the bond in-plane, cf. Fig. 4b, there are indications for a low frequency tail comparing to the magnetic lineshape. However, this is most pronounced at 18KBar and 300K, where also a similarly pronounced high-frequency tail is found that is not expected from angular distribution about Vyy. Finally, for cIIB, cf. Fig. 4a, a low frequency tail is discernible, but there is no indication of a high frequency tail that would also be expected for an angular distribution about Vxx.

*In summary, the O data cannot quantitatively rule out a small angular distribution about the crystal c-axis that the Cu data would allow for, but qualitatively there are no consistent indications for such a scenario.

Round 3

Reviewer 2 Report

The authors have answered on all my questions, and elaborated on all the doubts I had.